# Genome-Wide Identification of SRS Gene Family in Wheat and Expression Analysis Under Abiotic Stress

**DOI:** 10.3390/ijms26136289

**Published:** 2025-06-29

**Authors:** Yanan Yu, Qihang Chang, Chunyue Li, Kaiyue Wu, Yanyan Wang, Changhong Guo, Yongjun Shu, Yan Bai

**Affiliations:** 1Key Laboratory of Molecular Cytogenetics and Genetic Breeding of Heilongjiang Province, College of Life Science and Technology, Harbin Normal University, Harbin 150025, China; yynan2024@163.com (Y.Y.); changqh2024@163.com (Q.C.); licy2024@163.com (C.L.); wukaiyue1201@163.com (K.W.); kaku3008@hrbnu.edu.cn (C.G.); 2Department of Biomedical Engineering, School of Medical Devices, Shenyang Pharmaceutical University, Shenyang 110016, China; yyanan_07@163.com

**Keywords:** wheat, SRS gene family, bioinformatics, abiotic stress, expression analysis

## Abstract

The SHORT INTERNODES-related sequence (SRS) gene family, comprising zinc finger and IXGH domain-containing transcription factors, serves as a critical regulator of plant biological processes and abiotic stress responses. In this study, the common wheat cultivar Chinese Spring was selected as the experimental material. Comprehensive bioinformatic analysis was performed using ClustalX, MEGA, MEME, and PlantTFDB v5.0 to systematically characterize SRS family members within the wheat genome. The systematic examination of physicochemical properties, conserved domains, phylogenetic relationships, gene structures, and cis-acting elements was conducted, providing insights into the functional roles of this gene family in wheat growth and development. Fifteen SRS family members containing conserved zinc finger and IXGH domains were identified. Distinct expression patterns were observed among *TaSRS* subgroups: Members of Groups I, III, and V exhibited significantly higher transcript levels in roots, stems, leaves, and anthers compared to other subgroups. Notably, the majority of *TaSRS* genes, including representatives from Groups I, III, IV, and V, displayed responsiveness to NaCl and ABA stress treatments, suggesting their putative involvement in both salinity adaptation and phytohormone-mediated stress signaling. Differential expression patterns of *TaSRS* genes under NaCl and ABA stress were identified, revealing distinct regulatory impacts of these stressors on transcription. These findings establish a framework for investigating the molecular mechanisms underlying stress adaptation in wheat physiology.

## 1. Introduction

Transcription factors (TFs), functioning as trans-acting elements, are characterized by sequence-specific binding to cis-regulatory regions during transcriptional regulation. These proteins serve as critical regulators coordinating developmental processes and abiotic stress responses through modulation of gene expression networks [1]. The SHI-related sequence (SRS)-encoded protein is characterized by a conserved RING domain (CX_2_CX_7_CX_4_CX_2_C_2_X_6_C, X indicating variable amino acids) and an SHI family-specific IXGH domain [2,3]. The RING domain, comprising small protein motifs [4], contains a C3HC3H zinc finger structure that mediates interactions with nucleic acids, proteins, and lipids to regulate intracellular physiological and biochemical processes [5,6]. Within the IXGH domain, acidic amino acid residues are identified as functional transcriptional activation elements [7]. The SRS gene family has been extensively characterized in model plants including *Arabidopsis thaliana*, *Oryza sativa*, *Glycine max*, *Hordeum vulgare*, *Brassica rapa*, and *Malus domestica* [8]. Regulatory roles of SRS transcription factors have been established in organogenesis, photomorphogenesis, photoperiodic control, phytohormone biosynthesis, and abiotic stress responses [9,10,11]. The *Arabidopsis* genome encodes 11 *SRS* family members: *SHI*, *STYLISH1* (*STY1*), *STYLISH2* (*STY2*), *SRS3-8*, lateral root primordium 1 (*LRP1*), and an additional SHI homolog [12]. Functional analyses reveal *LRP1* involvement in auxin biosynthesis and chromatin remodeling during lateral root initiation [13]. *LRP1* regulates the expression of growth hormone synthesis gene *YUCCA4* [13]. SHI is characterized as a negative regulator within the gibberellin (GA) signaling pathway, suppressing premature developmental transitions through attenuation of GA-mediated responses [14]. *STY1* regulates auxin synthesis in *Arabidopsis* seedlings [15]. The coordinated regulation of shoot apical meristem establishment, stem differentiation, and floral organogenesis (particularly gynoecium and stamen development) is mediated through *STY1-SHI/STY* protein interaction, while concurrent modulation of leaf morphogenesis, cellular proliferation, and flowering time is achieved through this molecular partnership [16,17]. *SRS5* promotes seedling morphogenesis by activating photomorphogenetic genes (*HY5*, *BBX21*, and *BBX22*) [18]. Six *SRS* family members have been identified in *Oryza sativa*, with *OsSHI1* demonstrating synergistic interaction with the tillering regulator IPA1 to modulate plant architecture and yield parameters through the coordinated regulation of tiller development and panicle formation [19]. The functional characterization of the *SRS* gene family in *Chenopodium album* has demonstrated significant involvement in stress response pathways, with specific regulatory roles identified under multiple abiotic stress conditions [20]. The functional characterization of the *SRS* gene family in hexaploid wheat (*Triticum aestivum*) remains underexplored, representing a critical knowledge gap in understanding stress adaptation mechanisms within this agronomically vital crop species.

Saline and alkaline stresses are recognized as primary abiotic constraints severely restricting plant growth and development, posing substantial threats to global agricultural productivity and ecological sustainability [21]. Global agricultural lands are increasingly compromised by salinization, with approximately 20% of irrigated soils currently affected—a trend demonstrating progressive exacerbation. The dual stressors of salinity and alkalinity induce multifaceted phytotoxic effects, manifesting as osmotic imbalance, ion-specific toxicity, and salt-induced oxidative damage, while alkaline pH conditions generate secondary physiological impairments through nutrient availability disruption and cellular homeostasis interference [22]. Saline–alkaline stress is characterized by reduced soil osmotic potential, induced ionic imbalance, and impaired critical physiological functions, ultimately manifesting as growth inhibition, yield reduction, and phytotoxicity. Under extreme conditions, complete plant mortality may be observed. These cumulative effects have positioned plant adaptation mechanisms to saline–alkaline stress as a critical research priority within the global scientific community, particularly given its agricultural and ecological implications [23,24]. The present research priorities center on the identification of resistance genes, the comprehensive elucidation of underlying mechanisms, and the investigation of the molecular pathways through which plants respond to salt and alkali stress. These areas of study are still in their nascent stages. Specific focal points include the identification of salt- and alkali-tolerant genes, transcriptome sequencing of gene families, phylogenetic analysis, subcellular localization, and the utilization of protein interaction and expression profile data to facilitate in-depth predictions of gene function [25,26].

Abscisic acid (ABA) plays a crucial role in regulating various physiological processes throughout a plant’s life cycle [27]. These developmental and physiological processes—including seed maturation, leaf morphogenesis, stem cell homeostasis, stomatal regulation, photosynthetic efficiency, carbon partitioning, bud dormancy control, floral transition, fruit ripening coordination, source-sink dynamics, and senescence programming—are regulated through abscisic acid (ABA)-mediated signaling pathways. ABA is recognized as a key phytohormonal regulator in plant stress adaptation mechanisms, integrating environmental adversity responses with developmental plasticity [28,29]. The perception and transduction of stress signals is mediated by ABA, triggering coordinated cellular responses that augment plant resilience to multiple abiotic stressors [30]. The elucidation of ABA-mediated physiological mechanisms under abiotic stress conditions has been recognized as critical for advancing understanding of plant adaptation strategies and optimizing agronomic applications in stress mitigation. This domain has consistently maintained its position as a central research focus within plant stress physiology, bridging fundamental biological discoveries with practical agricultural innovations [31,32].

Wheat (*Triticum aestivum* L.), a herbaceous plant belonging to the Gramineae family and Triticum genus, is either annual or biennial [33]. Over time, it has evolved to become one of the most widely distributed, extensively cultivated, second highest in total production, most traded, and nutritionally valuable food crops globally [34]. Wheat cultivars are systematically classified into two agronomic types by sowing season: spring wheat (*Triticum aestivum* ssp. *vernale*) and winter wheat (*Triticum aestivum* ssp. *hybernum*). Spring wheat is characterized by vernalization-independent growth cycles, with sowing initiated in early spring (March–April) and harvest completed within the same growing season (July–September). Conversely, winter wheat requires vernalization exposure, typically sown in autumn (September–November) to establish root systems prior to winter dormancy, with grain maturation achieved by early summer (June–July) of the subsequent year [35]. Wheat (*Triticum aestivum* L.) is cultivated as a staple crop across approximately 33% of global arable land, representing the predominant cereal species allocated for human dietary consumption worldwide [36,37]. Wheat yield and quality are significantly influenced by multiple abiotic stressors throughout developmental stages, particularly drought, salinity, and low-temperature conditions [38]. Under abiotic stress conditions, transcriptional reprogramming is mediated through SRS transcription factors, which serve as critical regulatory components in orchestrating stress-responsive signaling networks and initiating adaptive gene expression modifications.

A genome-wide identification of the *SRS* gene family in wheat was systematically conducted, encompassing chromosomal localization mapping, physicochemical property characterization, conserved motif analysis, phylogenetic reconstruction, and spatiotemporal expression profiling. The expression patterns across developmental tissues and stress-responsive regulatory mechanisms were comprehensively characterized, establishing a molecular foundation for targeted genetic improvement of stress-resilient wheat cultivars through precision breeding strategies [39,40].

## 2. Results

### 2.1. Identification and Physicochemical Properties of Wheat SRS Genes

A systematic genome-wide analysis of wheat (*Triticum aestivum*) was conducted using integrated bioinformatic tools (Pfam, HMMER, CDD, SMART), identifying 15 *SRS* family members designated *TaSRS1*-*TaSRS15* based on chromosomal coordinates (Table 1). The compiled genomic dataset characterizes key physicochemical parameters including systematic nomenclature, predicted isoelectric points (pI 7.70–8.96), exon–intron architecture (2–3 exons per gene), and protein length variations (247–352 residues). Notably, *TaSRS5* encodes the longest polypeptide (352 residues), while *TaSRS4* produces the shortest protein (247 residues), representing the extreme length variations within this gene family.

### 2.2. Analysis of Conserved Motif and Phylogenetic of Wheat SRS Gene Family

To characterize the structural features of SRS family genes, conserved protein motifs were systematically analyzed using the MEME suite through cross-species comparative genomics (Figure 1a). The 78 analyzed SRS homologs exhibited motif diversity ranging from one to nine conserved domains, with wheat TaSRS proteins demonstrating higher structural complexity (five–nine motifs). Notably, TaSRS4 contained the minimal motif complement (five domains). Multiple sequence alignment confirmed universal conservation of two signature domains across all SRS proteins: a zinc finger structure (designated Motif 1) and an IXGH domain (identified as Motif 2), with respective positional conservation illustrated in Figure 1b.

To elucidate the phylogenetic relationships of TaSRS proteins, an unrooted neighbor-joining (NJ) tree was constructed using 15 protein sequences from *Triticum aestivum* and 6 sequences from *Oryza sativa* (Figure 2a). Based on the tree topology and consistency with the OsSRS classification, the 21 SRS proteins from these species were classified into five major subfamilies (Figure 2), with the groupings described throughout the text corresponding to Figure 2a. Group I contains three wheat proteins (TaSRS7, TaSRS8, TaSRS9) and one rice protein (OsSRS6); Group II contains three wheat proteins (TaSRS10, TaSRS12, TaSRS14) and one rice protein (OsSRS5); Group III contains three wheat proteins (TaSRS11, TaSRS13, TaSRS15) and one rice protein (OsSRS4); Group IV contains three wheat proteins (TaSRS4, TaSRS5, TaSRS6) and two rice proteins (OsSRS1, OsSRS2); Group V contains three wheat proteins (TaSRS1, TaSRS2, TaSRS3) and one rice protein (OsSRS3). Bootstrap values ≥ 97% were obtained for all wheat and rice protein classifications within each group, indicating robust support for the assignments. The phylogeny clearly demonstrates a consistent distribution pattern of SRS proteins between wheat and rice.

To investigate the evolutionary relationships of *TaSRS* genes across species, an unrooted neighbor-joining (NJ) tree was constructed using sequences from monocots (*Triticum aestivum*, *Oryza sativa*, *Zea mays*) and dicots (*Glycine max*, *Brassica rapa*, *Arabidopsis thaliana*), elucidating the *TaSRS* gene family’s evolution between monocotyledonous and dicotyledonous plants (Figure 2b). Based on topological consistency, 78 SRS proteins were classified into four major subfamilies (Figure 2b). Group IV contained the highest number of proteins, while Group I contained the fewest. Group I comprised eight proteins (four soybeans, two cabbages, two Arabidopsis; all dicots). Group II included 23 proteins (13 dicots, 10 monocots) with distinct clustering by plant type. Group III consisted exclusively of 15 dicot proteins. Group IV contained 15 monocot and 15 dicot proteins, each forming separate clades. This robust classification revealed significant evolutionary divergence: phylogenetic analysis demonstrated that wheat *TaSRS* genes share closest homology with rice and maize, indicating recent common ancestry, while exhibiting greater genetic distance from dicot species.

### 2.3. Analysis of Wheat SRS Gene Structure and Prediction of Protein Tertiary Structure

The structure of genes plays a crucial role in the evolution of gene families. To better understand gene structure, an intron–exon model of the *TaSRS* genome sequence was constructed. As shown in Figure 3, the results are consistent with the phylogenetic tree, indicating that the intron–exon structural models within the same subfamily are similar, yet not identical. This may be due to variations in the function of *SRS* genes during the evolutionary process. The number of exons in members of the *TaSRS* gene family ranges from two to three, while the number of introns ranges from one to two. In Group I, all members have two exons and one intron. In Group II, *TaSRS10* and *TaSRS14* have two exons and one intron, while *TaSRS12* has three exons and two introns. Members of Group III have three exons and two introns. In Group IV, *TaSRS4* has two exons and one intron, while *TaSRS5* and *TaSRS6* have two exons and two introns. Members of group V have two exons and two introns.

To elucidate the tertiary structure of wheat SRS proteins, predictions were conducted using an online platform. Based on the predictions from SWISS-MODEL (https://swissmodel.expasy.org, accessed on 5 June 2025), the three-dimensional structures of 15 TaSRS proteins were modeled (Figure 4a). Additionally, the conserved domain regions within these 15 proteins were also analyzed for their three-dimensional structures (Figure 4b). The results demonstrated that the three-dimensional structures of TaSRS proteins were successfully predicted, with all proteins exhibiting α-helix and β-sheet structures. Members of the same family displayed similar structural features. This structural analysis facilitates a deeper understanding of their functional characteristics.

### 2.4. Analysis of Chromosome Localization and Collinearity of Wheat SRS Gene

As shown in Figure 5, the chromosomal localization of *TaSRS* revealed that the 15 members of the *SRS* family in wheat are distributed across 12 chromosomes. Specifically, chromosomes 1A, 1B, 1D, 3A, 3B, 3D, 5A, 5B, and 5D each contain a single *SRS* gene, while chromosomes 7A, 7B, and 7D harbor two SRS genes each.

To elucidate the evolutionary origins of the *SRS* gene family in wheat, a homology analysis was conducted on *SRS* family members from wheat and rice. The tandem duplication patterns are illustrated in Figure 6, where red lines highlight conserved tandem *SRS* gene pairs between the two species, and gray lines denote collinear blocks. Identified tandem gene pairs revealed conserved arrangements in both genomes, including 12 wheat *SRS* genes and 4 rice *SRS* genes. Notably, all wheat *SRS* family members exhibited corresponding orthologs in the rice *SRS* family. This finding underscores a shared evolutionary history between wheat and rice, suggesting significant conservation of the *SRS* gene family across these species.

### 2.5. Analysis of Cis-Acting Elements of Wheat SRS Gene Family

To gain a deeper understanding of the regulatory mechanisms and potential functions of the *TaSRS* gene, the cis-acting elements within its promoter sequence were analyzed using the online tool PlantCARE (https://bioinformatics.psb.ugent.be/webtools/plantcare/html/, accessed on 25 June 2025) (Figure 7). In addition to the core promoter elements, a total of 15 cis-acting elements were identified (Figure 7). As illustrated in the figure, as depicted in the figure, among these, there are five elements related to plant hormones: the abscisic acid response element (ABRE), the methyl jasmonate response elements (CGTCA-motif and TGACG-motif), the gibberellin response element (P-box), and the auxin response element (TGA-element). One element associated with growth and development was identified, namely the meristem expression element (CAT-box). There are five light-responsive elements: Box 4, GATA-motif, G-box, TCCC-motif, and Sp1. Four stress-related elements were found: the drought stress response element (MBS), anaerobic induction elements (ARE and GC-motif), and the low-temperature response element (LTR). Among these, ABRE, CGTCA-motif, and G-box are the most abundant within their respective categories of hormone-related, stress-related, and light-responsive elements.

### 2.6. Analysis of the Gene Regulatory Network of the TaSRS Gene Family in Wheat

Protein interaction information with known proteins can be identified through the STRING database (STRING: functional protein association networks), which provides a better understanding of the complex regulatory networks within biological systems. Analysis of these networks reveals that the majority of *TaSRS* genes interact with several functional genes, highlighting their roles in transcriptional regulation (Figure 8a). For example, *TaSRS8* (AOA3B6LQX1) interacts with six genes, *TaSRS14* (A0A3B6TRD6) with seven genes, and *TaSRS15* (A0A3B6TVV4) with three genes, emphasizing the significance of these *TaSRS* genes.

To further elucidate the functions of these interacting genes, gene KEGG pathway enrichment was examined (Figure 8b). The significance of the pathways is indicated by the false discovery rate (FDR), with lower values corresponding to higher significance. Notably, significant pathways include the biosynthesis of phenylalanine, tyrosine, and tryptophan. These amino acids serve as precursors for secondary metabolites such as phenolic compounds and lignin, enhancing plant antioxidant capacity and cell wall resistance to pathogens and participating in abiotic stress responses (drought and high salinity). The gene count in these pathways ranges from 5 (few genes enriched) to 90 (many genes enriched). Pathways with a higher number of genes are typically involved in broadly participating cellular metabolic processes, such as porphyrin and chlorophyll metabolism. Light inhibition leads to chlorophyll degradation, while porphyrin metabolites (heme) are involved in reactive oxygen species (ROS) signaling regulation and participate in responses to light and heavy metal stresses. Pathways with fewer genes may represent more specific functions. Pathways that are both highly significant and have a high number of genes are likely to be core pathways of interest. These include metabolic pathways (carbon metabolism, amino acid synthesis) and signaling pathways (MAPK). Amino acid synthesis contributes to osmotic balance maintenance through the accumulation of compounds such as proline and glycine betaine, participating in responses to drought, high salinity, and low-temperature stresses. The MAPK pathway phosphorylates transcription factors such as WRKY, MYB, and NAC, driving the expression of stress-related genes and interacting with plant hormone signaling (abscisic acid [ABA], jasmonic acid [JA], salicylic acid [SA]) to form a coordinated defense network. These analyses reinforce the known functions of *TaSRS* genes and provide a broader functional context for *TaSRS* within the wheat SRS gene family.

### 2.7. The Expression Profile of the Wheat TaSRS Gene in Response to Abiotic Stress

To further elucidate the role of *TaSRS* in response to abiotic stress, RNA-seq data were collected from wheat plants subjected to various stress conditions. RNA sequencing was performed on wheat plants under low temperature, salt, drought, and ABA stress conditions. Visualization of the results revealed that, except for gene *TaSRS11*, the expression of other genes was relatively low under cold treatment (Figure 9a). Under drought treatment, the expression of all genes except *TaSRS3* and *TaSRS15* was weak (Figure 9b). Most *TaSRS* genes were upregulated in response to salt and ABA stress (Figure 9c,d). Under salt stress, the expression patterns of 10 genes were upregulated, with *TaSRS1*, *TaSRS2*, *TaSRS7*, *TaSRS11*, and *TaSRS13* showing particularly high expression levels. Under ABA stress, the expression patterns of 12 genes were upregulated, with *TaSRS2*, *TaSRS3*, *TaSRS6*, and *TaSRS8* exhibiting particularly high expression levels. These results strongly suggest that TaSRS genes play a crucial role under ABA and salt stress conditions.

### 2.8. Expression of Wheat SRS Gene Family Under Different Tissue, Abiotic Stress and Hormone Treatment

Previous studies have elucidated the role of the SRS gene in plant growth and development, including root formation, plant stature, and hormone synthesis. The expression patterns of *TaSRS* across various tissues (roots, stems, leaves, and anthers) were examined using quantitative real-time polymerase chain reaction (qRT-PCR). As illustrated in Figure 10a, the SRS gene was detected in all the aforementioned tissues. Notably, Families I, III, and V exhibited higher expression levels in roots, stems, leaves, and anthers compared to other members, with Family III displaying the highest expression. In contrast, Families II and IV demonstrated relatively lower expression levels in these tissues.

Plant growth and development are severely impacted by stress. Therefore, understanding the expression profile of the *TaSRS* gene under different stress treatments can help reveal the potential functions of these genes. To further validate the pronounced responses of *TaSRS* to abscisic acid (ABA) and salt stress. The expression patterns of the *TaSRS* gene under cold, salt, ABA, and drought stress were analyzed using qRT-PCR technology (Figure 10b). The results indicate that under various abiotic stresses, the majority of genes responded to ABA and salt stresses, albeit at different expression levels. Subfamilies I, III, IV, and V were significantly upregulated in response to salt and ABA stresses, with Subfamily III also responding to cold and drought stresses. Notably, Subfamily III exhibited an approximate fourfold upregulation in response to salt stress, while Subfamily I displayed a nearly fourfold upregulation in response to ABA stress. These findings provide compelling evidence that *TaSRS* is actively involved in responding to ABA and salt stress conditions and plays a crucial role in regulating gene expression in wheat.

## 3. Discussion

Transcription factors play a pivotal role in plants’ responses to various biotic and abiotic stresses [41]. The *SRS* family of transcription factors, ubiquitous in eukaryotes, possesses a distinctive zinc finger motif (CX_2_CX_7_CX_4_CX_2_C_2_X_6_C, where X represents variable amino acids) and an IXGH domain [42]. To date, the *SRS* family has been identified across multiple species, including *Arabidopsis thaliana*, *Oryza sativa* [43], *Glycine max*, *Hordeum vulgare*, *Brassica rapa*, and *Malus domestica*. Studies on these plants have revealed that most amino acid sites within the zinc finger motif and IXGH domain of SRS members are highly conserved. In Arabidopsis, *SRS* family genes have been shown to be involved in the regulation of photomorphogenesis, organ development, lateral root formation, carpel development, and plant hormone synthesis [16,44]. Among them, *AtLRP1*, as the first reported SRS gene, regulates root system development by modulating auxin and histone deacetylation [13,45]. In tomato, *SlSRS7* is upregulated by auxin and predominantly expressed in the root system, while *SlSRS8* is mainly expressed in floral organs [10,46]. Studies have indicated that *OsSHI1* in rice and some *AtSRS* genes are highly expressed in roots but lowly in leaves [9,17,47]. Multiple studies suggest that the SRS family participates in various complex regulatory processes during growth and development [48]. However, there have been no reports on the SRS family in wheat. Therefore, this study conducted a genome-wide analysis of the *TaSRS* family.

In order to analyze the physicochemical characteristics of various genes in the wheat SRS family, a total of 15 *TaSRS* family members were identified in this study, which is more than the family members of *Arabidopsis thaliana*, *rice*, *maize*, *tomato*, and *cassava*, but less than the family members of *upland cotton* [49]. This indicates that the SRS gene family has undergone amplification or reduction during the long-term species evolution. The physicochemical properties of *TaSRS* family are similar to those of cassava, tomato, and rice, with amino acid numbers ranging from 200 to 400 aa [43]. The pI values are all in the range of six to eight, and the proteins are all slightly alkaline. However, there are differences in the amino acid numbers, molecular weights, and isoelectric points among the *TaSRS* family members. For example, *TaSRS7*, *TaSRS8*, and *TaSRS9* have the largest pI values, while *TaSRS6* has the smallest pI value. This may be closely related to the diversity of *TaSRS* structure and biological function.

A phylogenetic analysis was conducted on the SRS proteins of rice and wheat, leading to the classification of wheat and rice *SRS* members into five subfamilies (Subfamily I, II, III, IV, V) based on the structure and type of conserved motifs [50]. In rice, *OsSRS1* and *OsSRS2* are found in Group IV, with each of the other subfamilies containing a single *OsSRS* member, consistent with previous reports [51]. *TaSRS* is evenly distributed across the five subfamilies, with three members in each. The gene and protein structures of wheat and rice SRS members exhibit significant similarity, suggesting that the SRS proteins of wheat and rice may share analogous functions. Furthermore, to explore the diversity of SRS gene structures in wheat, an analysis of wheat SRS gene structures was performed. The results indicated that most SRS genes within the same subfamily possess an identical number of exons and introns, implying that genes within the same subfamily might have similar functions [52].

Cis-acting regulatory elements are non-coding DNA segments identified within the promoter regions of genes [53]. An analysis of these cis-elements in members of the *TaSRS* gene family has revealed numerous elements associated with phytohormones and abiotic stress responses, including abscisic acid (ABA), ethylene, gibberellins, auxins, low temperatures, and drought. Among these, ABA-responsive element binding proteins (AREBs/ABFs) can bind to the ABA-responsive element (ABRE), participating in reactions to ABA, dehydration, and high salt stress. The ABRE is present in the promoter regions of species such as wheat, rice, quinoa, tomato, apple, and Chinese cabbage. This indicates that the *TaSRS* gene plays a significant role in both biotic and abiotic stress responses [54].

Expression profiling of 15 *TaSRS* family members via qRT-PCR revealed that wheat *SRS* genes are expressed across various tissues, albeit to varying degrees. Notably, the majority of genes responded to ABA and salt stresses, albeit at different expression levels. Subfamilies I, III, IV, and V were significantly upregulated in response to salt and ABA stresses, with Subfamily III also responding to cold and drought stresses, indicating tissue-specific expression patterns during wheat development. Within the *melilotus albus*, *MaSRS09* is significantly upregulated under salt stress at 6 h and 24 h stress time points. Within a 48 h cold treatment period, the expression levels of *MaSRS07* and *MaSRS09* are significantly elevated [20]. In *Brassica napus*, the *BnSRS18* gene consistently exhibited upregulation under various abiotic stresses, with the highest expression observed after 24 h of salt treatment, showing an increase of nearly twofold. [5]. In the quinoa species, all SRS genes respond to SA, NaCl, and cold treatments. Under NaCl and cold treatments, most genes exhibit similar expression patterns, with post-treatment expression levels significantly exceeding those of the control group [7]. In soybeans, the expression of *GmSRS18* and *GmSRS21* increases under ABA treatment and decreases under drought conditions [3]. Additionally, in apples, the expression levels of *MdSRS2*, *MdSRS5*, *MdSRS8*, and *MdSRS9* exhibit a declining trend as the duration of drought stress extends. Conversely, *MdSRS4* demonstrates an initial decrease followed by an increase. Under ABA treatment, *MdSRS* genes, exemplified by *MdSRS2*, show an initial upregulation followed by a downregulation, though the time point at which peak expression is reached varies. The *MdSRS3* gene consistently shows an increase in expression. while a subset of these genes show upregulation in response to ABA stress. In summary, various SRS genes exhibit distinct expression levels under salt and ABA stresses. This indicates that salt and ABA exert differential effects on the expression of *TaSRS* genes, suggesting that *TaSRS* genes play a role in the specific regulation of plant hormones during wheat development.

In summary, 15 wheat SRS gene family members were identified in this study, and their physical and chemical properties, phylogeny, gene structure, chromosome localization, cis-acting elements and expression patterns were analyzed, providing a theoretical basis for further exploration of this gene family.

## 4. Materials and Methods

### 4.1. Plant Material

The common wheat cultivar ‘Chinese Spring’ (*Triticum aestivum* L.) was selected for this study. Plump seeds were surface sterilized with 15% (*v*/*v*) sodium hypochlorite for 5 min, followed by three rinses in sterile distilled water. Seeds were evenly distributed in sterile Petri dishes and germinated at 28 °C for 24 h in darkness. Uniformly sprouted seedlings were transferred to hydroponic growth chambers containing 1/2 Hoagland nutrient solution and maintained in a controlled greenhouse (16 h light/8 h dark photoperiod, constant temperature). For tissue-specific expression analysis, roots, stems, and leaves were harvested from 14-day-old seedlings. To assess stress responses, seedlings at the same developmental stage were subjected to abiotic treatments: 0.2 M NaCl (salt stress), 20% PEG-6000 (osmotic stress), 0.1 mM abscisic acid (ABA), or 4 °C (cold stress) for 6 h. Untreated seedlings grown under standard conditions served as controls. All tissues (roots, stems, leaves) from both treated and control groups were collected in triplicate, immediately flash-frozen in liquid nitrogen, wrapped in pre-chilled aluminum foil, and stored at −80 °C for subsequent RNA extraction. Pollen samples collected from field-grown plants were similarly processed in triplicate.

### 4.2. Identification of Wheat SRS Gene Family Members

To identify members of the SRS gene family in wheat (*Triticum aestivum*), the whole-genome sequence of wheat was retrieved from the Ensembl Plants database (http://plants.ensembl.org, accessed on 25 June 2025). The Hidden Markov Model (HMM) profile corresponding to the SRS domain (PF01542) was obtained from the Pfam database (http://pfam.xfam.org, accessed on 25 June 2025). Candidate protein sequences were initially screened using HMMER (version 3.0) with the SRS domain HMM as a query, followed by domain validation via the SMART (http://smart.embl.de, accessed on 25 June 2025) and NCBI Conserved Domain Database (CDD) tools (https://www.ncbi.nlm.nih.gov/cdd/, accessed on 25 June 2025). Sequences lacking either the zinc finger or IXGH domain were excluded, resulting in a non-redundant set of wheat SRS proteins. The amino acid sequence of wheat *TaSRS* was analyzed for its length, isoelectric point, molecular weight, and other physicochemical properties using the ExPASy website (https://web.expasy.org/protparam/, accessed on 25 June 2025). Predicted domain sequences were obtained from the PlantTFDB v5.0 website (http://planttfdb.cbi.pku.edu.cn/, accessed on 25 June 2025) (version v5.0).

### 4.3. Analysis of the Conserved Motifs and Phylogenetic Evolution of the SRS Gene Family

Conserved motifs within the SRS gene family were predicted for maize (*Zea mays*), wheat (*Triticum aestivum*), rice (*Oryza sativa*), soybean (*Glycine max*), Arabidopsis (*Arabidopsis thaliana*), and cabbage (*Brassica oleracea*) using MEME Suite (v4.8.1; https://meme-suite.org, accessed on 25 June 2025), with the maximum motif number set to 10. SRS protein sequences were retrieved from the PlantTFDB v5.0 database (http://planttfdb.cbi.pku.edu.cn, accessed on 25 June 2025). Multiple sequence alignment of SRS amino acid sequences was performed with ClustalX, followed by phylogenetic tree construction in MEGA11.0 using the Neighbor-Joining method (Poisson correction, pairwise deletion, 1000 bootstrap replicates).

### 4.4. Analysis of the Structure of Wheat SRS Gene and Three-Dimensional Structure of Wheat SRS Protein

Gene structure and chromosomal localization data for the wheat SRS gene family were extracted from the Triticum aestivum IWGSC RefSeq v2.1 genome annotation (GFF3 file). Intron–exon architectures were visualized using the Gene Structure Display Server (GSDS; http://gsds.cbi.pku.edu.cn, accessed on 25 June 2025). Tertiary protein structures of *TaSRS* members were predicted via homology modeling on the SWISS-MODEL platform (https://swissmodel.expasy.org, accessed on 25 June 2025), with templates selected based on sequence identity (>70%).

### 4.5. Analysis of Chromosome Localization and Synteny of the SRS Gene

The chromosome location information was visualized by the online website MG2C (http://mg2c.iask.in/mg2c_v2.1/, accessed on 25 June 2025). Tbtools software (version 1.098) was used to draw the colinearity relationship of wheat and rice gene families.

### 4.6. Analysis of Cis-Acting Elements of SRS Gene Family in Wheat

The DNA sequences of the 2000 bp upstream region from the promoter of the wheat SRS gene were obtained from the whole-genome database of wheat. The Plant CARE software (https://bioinformatics.psb.ugent.be/webtools/plantcare/html/, accessed on 25 June 2025) was utilized to analyze cis-acting elements related to plant hormone and stress responses.

### 4.7. TaSRS Gene Family Regulatory Network Analysis in Wheat

To evaluate the interactions of *TaSRS* proteins with other interactors, the protein sequences of *TaSRS* were utilized in the STRING database (STRING: functional protein association networks) (https://cn.string-db.org/, accessed on 25 June 2025) to construct an interaction network. The interactive network illustrates the interconnections between the query proteins and their interactors, with some being interconnected. Additionally, gene KEGG pathway enrichment was generated to further elucidate the functional context of these interactions.

### 4.8. The Expression Profile of the Wheat TaSRS Gene in Response to Abiotic Stress

RNA-seq samples from four different abiotic stress conditions were downloaded from the NCBI Sequence Read Archive database (http://www.ncbi.nlm.nih.gov/sra, accessed on 25 June 2025). These samples are publicly available under the following accession numbers: low temperature (control: SRR1460549, SRR1460550; stress: SRR1460552, SRR1460553), drought (control: SRR1542404, SRR1542405; stress: SRR1542408, SRR1542409), salt (control: SRR7920873, SRR7975953; stress: SRR7968053, SRR7968059), and ABA (control: SRX4062828, SRX4062839; stress: SRX4062829, SRX4062836). These data were utilized to investigate the expression patterns of *TaSRS*. Quality control and filtering were performed using FastQC and Trim Galore, respectively. The reads were aligned to the wheat reference genome using HISAT2 (version 2.2.1). Subsequently, the expression levels of *TaSRS* were calculated and normalized using Cufflinks. A heatmap was generated using the R package (version 4.3.2) “pheatmap” (version 1.0.12) to visualize the expression patterns of *TaSRS* genes under different conditions.

### 4.9. Quantitative Expression Analysis of SRS Gene Family in Wheat

Total RNA extraction was extracted from wheat using the RNApure Plant Kit (Tiangen, Beijing, China). Subsequently, cDNA synthesis was conducted using the Prime Script RT kit (Toyobo, Shanghai, China) to serve as a template for the quantitative reverse transcription PCR (qRT-PCR). Primers were designed for specific fragments of each subfamily, and their specificity was confirmed through agarose gel electrophoresis followed by sequencing alignment (Table 2). The qRT-PCR conditions and reaction systems followed the protocol for SYBR Premix Ex TaqTMII (Toyobo, Shanghai, China). The PCR reaction mixture consisted of the following components: 12 μL of cDNA, 10 μL of 2× TransStart^®^ Top Green qPCR SuperMix (Transgene, Beijing, China), 0.4 μL each of forward and reverse primers, and 7.2 μL of Nuclease-free Water. qRT-PCR analysis was conducted with three replicates of each experiment. Data acquisition was executed using the LightCycler^®^ 96 system (Roche, Basel, Switzerland). Fold change values were calculated based on gene expression abundance, employing the 2^−∆∆CT^ method.

## Figures and Tables

**Figure 1 ijms-26-06289-f001:**
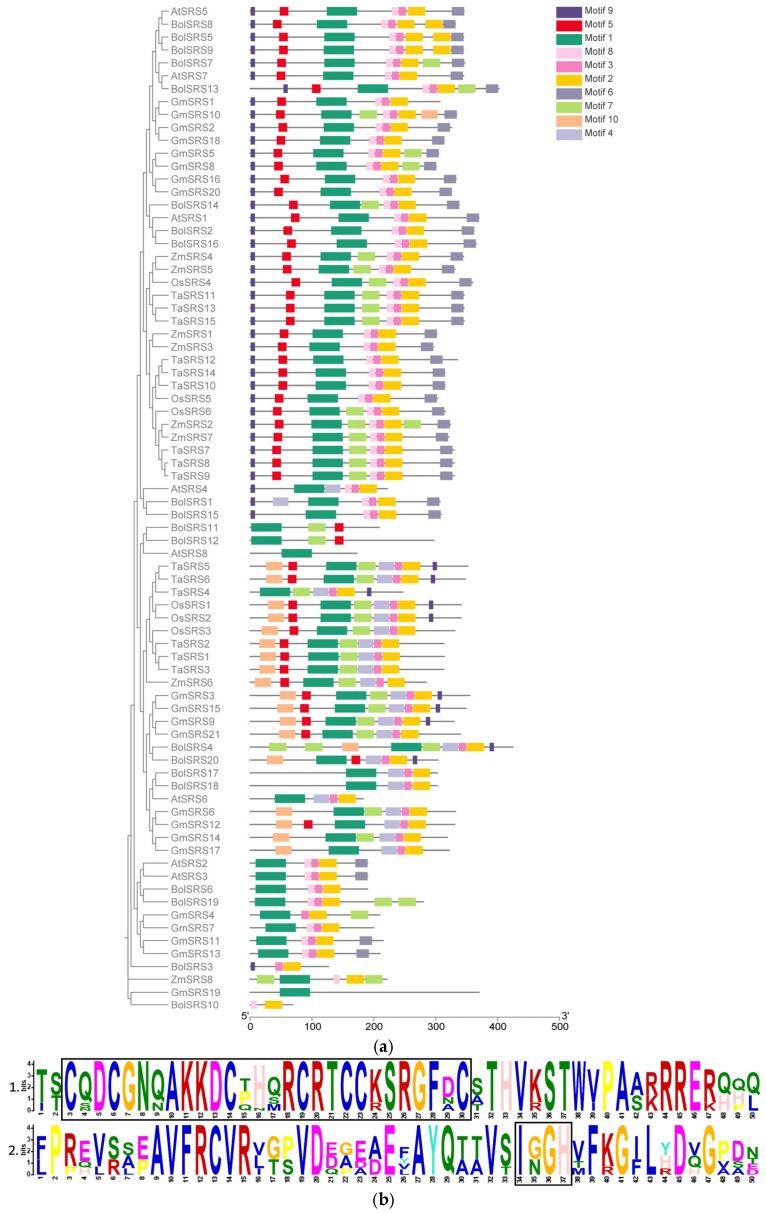
Conserved motif distribution in SRS proteins was systematically analyzed across six species: maize (*Zea mays*), wheat (*Triticum aestivum*), rice (*Oryza sativa*), soybean (*Glycine max*), cabbage (*Brassica rapa*), and *Arabidopsis thaliana*. (**a**) indicates the conserved domain of the SRS gene family. (**b**) Motif 1, corresponding to zinc finger (CX2 CX7 CX4 CX2 CX7 C). Motif 2, corresponding to IXGH domain Branch, black boxes annotate the motifs. Numerical values 1 and 2 correspond to motif ranking positions as specified in the analysis. Motif 1 is represented by the green structural domain in (**a**), while Motif 2 corresponds to the yellow conserved region within the schematic.

**Figure 2 ijms-26-06289-f002:**
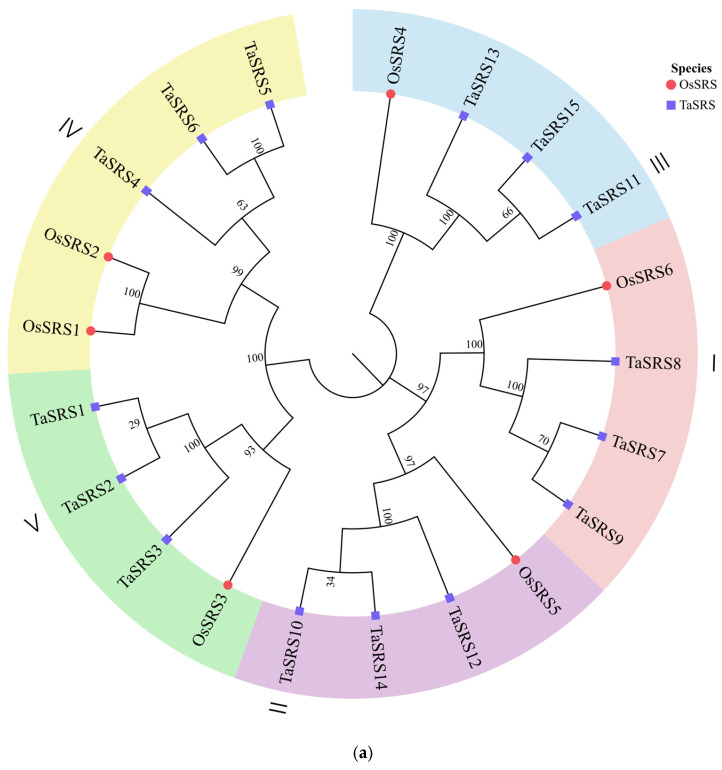
Phylogenetic tree of SRS proteins. The NJ tree is made of the TaSRS amino acid sequence using MEGA11 with 1000 bootstrap repeats. (**a**) depicts the phylogenetic relationships of the SRS gene family within monocotyledonous species *Triticum aestivum* (wheat) and *Oryza sativa* (rice), while (**b**) illustrates evolutionary relationships across monocots (wheat, rice, *Zea mays*) and dicots (*Arabidopsis thaliana*, *Glycine max*, *Brassica rapa*).

**Figure 3 ijms-26-06289-f003:**
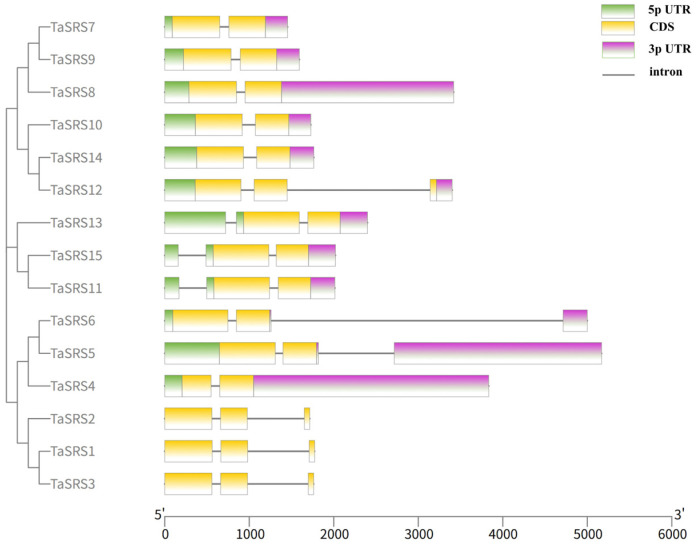
Gene structure of SRS gene family in wheat. The green color is utilized to denote the 5′ untranslated region (5′ UTR), while yellow indicates coding sequences (CDS). Horizontal lines are employed to represent introns, and pink shading specifies the 3′ untranslated region (3′ UTR).

**Figure 4 ijms-26-06289-f004:**
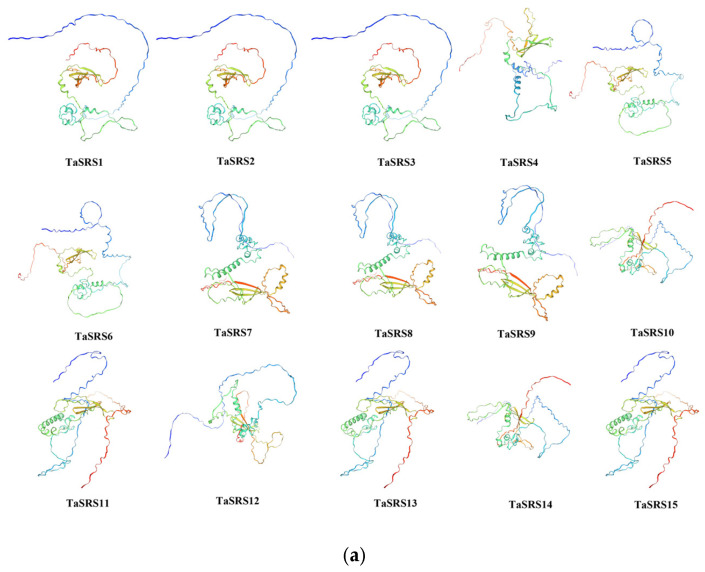
Tertiary structure of TaSRS protein. (**a**) “represents the tertiary structure of TaSRS protein”. α-helix (green), irregular coil (blue and red). (**b**) “represents the tertiary structure of the TaSRS protein domain”. α-helix (green), irregular coil (blue and red).

**Figure 5 ijms-26-06289-f005:**
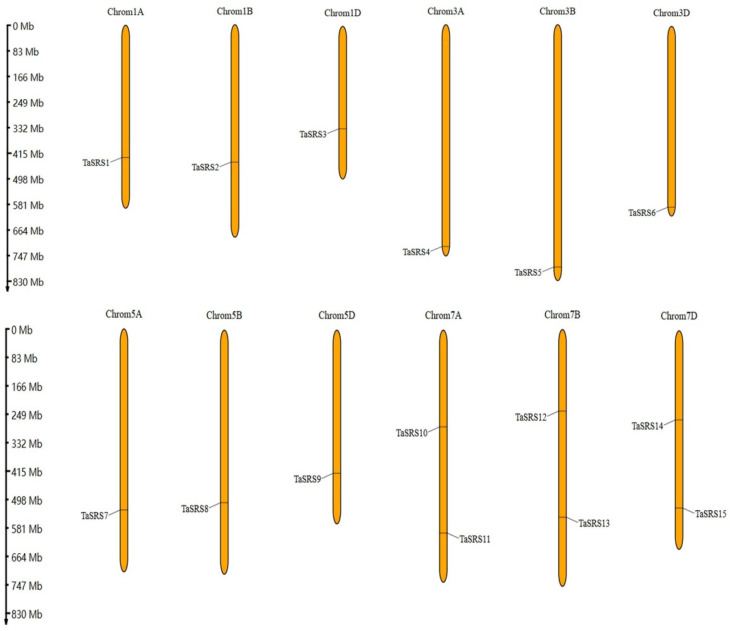
The chromosome location of the *SRS* gene family in wheat. *TaSRS1*, *TaSRS2*, and *TaSRS3* were mapped to chromosomes 1A, 1B, and 1D, respectively; *TaSRS4* and *TaSRS5* were assigned to chromosomes 3A and 3B; *TaSRS6* was positioned on chromosome 3D; *TaSRS7* and *TaSRS8* were localized to chromosomes 5A and 5B; *TaSRS9* was identified on chromosome 5D; *TaSRS10* and *TaSRS11* were co-localized on chromosome 7A; *TaSRS12* and *TaSRS13* were jointly mapped to chromosome 7B; while *TaSRS14* and *TaSRS15* were concurrently positioned on chromosome 7D.

**Figure 6 ijms-26-06289-f006:**
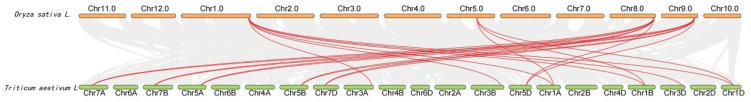
Collinearity analysis of *SRS* between wheat and rice. The collinear gene pairs with *TaSRS* are highlighted through the red lines, while the collinear blocks are marked through gray lines.

**Figure 7 ijms-26-06289-f007:**
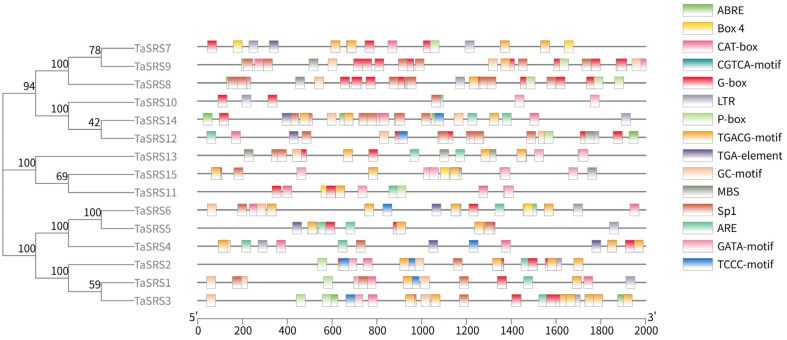
Cis-acting element analysis of *TaSRS*. ABRE: cis-acting element involved in the abscisic acid responsiveness; Box 4: part of a conserved DNA module involved in light responsiveness; CAT-box: cis-acting regulatory element related to meristem expression; CGTCA-motif: cis-acting regulatory element involved in the MeJA-responsiveness; G-Box: cis-acting regulatory element involved in light responsiveness; LTR: cis-acting element involved in low-temperature responsiveness; P-box: gibberellin-responsive element; TGACG-motif: cis-acting regulatory element involved in the MeJA-responsiveness; TGA-element: auxin-responsive element; GC-motif: enhancer-like element involved in anoxic specific inducibility; MBS: MYB binding site involved in drought-inducibility; Sp1: light responsive element; ARE: cis-acting regulatory element essential for the anaerobic induction; GATA-motif: part of a light responsive element; TCCC-motif: part of a light responsive element.

**Figure 8 ijms-26-06289-f008:**
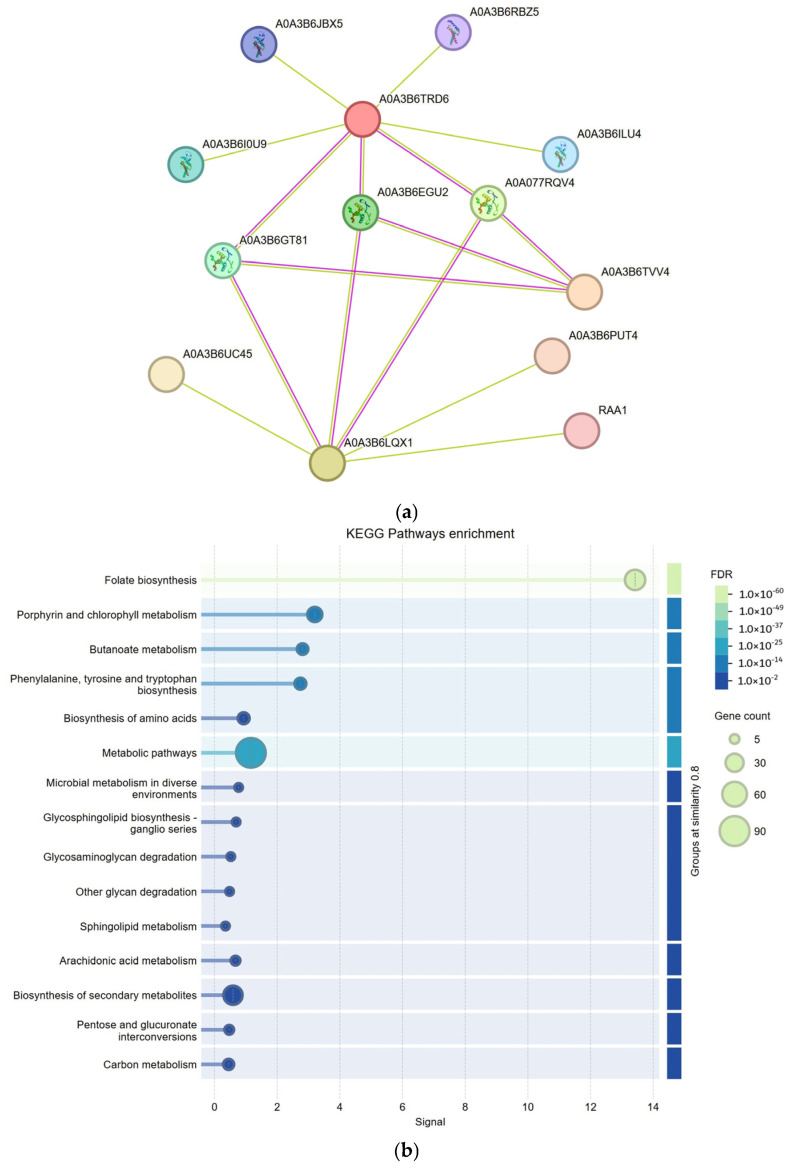
The gene regulatory network of the *TaSRS* gene family in wheat. Network Diagram (**a**) illustrates the protein interaction network. Nodes represent individual proteins, with node labels indicating the names of these proteins. The patterns within the nodes depict the three-dimensional structures of the proteins; an empty pattern indicates that the structure is currently unknown. Interactions between proteins are indicated by connecting lines, with the color of the line reflecting the type of interaction: yellow lines signify text-mining evidence, while purple lines indicate experimentally validated interactions. In Network Diagram (**b**), KEGG pathway enrichment is represented. The False Discovery Rate (FDR) indicates the adjusted significance level, while the Gene Count represents the number of genes enriched in a particular pathway. The FDR values range from 1.0 × 10^−6^ (highly significant) to 1.0 × 10^−2^ (relatively less significant), indicating that the pathways depicted in the figure span from highly significant to moderately significant.

**Figure 9 ijms-26-06289-f009:**
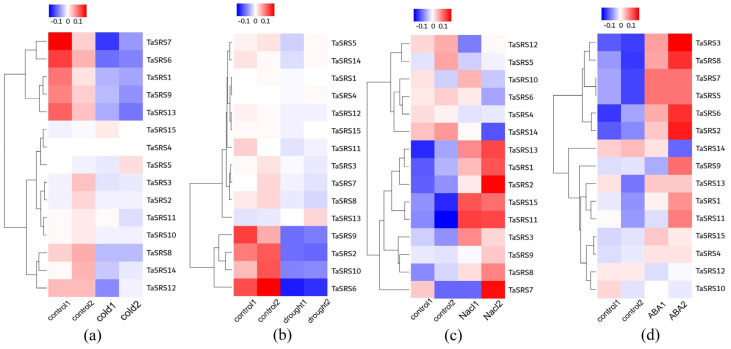
The expression profile of the *TaSRS* gene family under abiotic stress. Red indicates high expression and blue indicates low expression. The X-axis represents the control and stress samples. The Y-axis represents the 15 TaSRS genes. (**a**) represents the low-temperature group, (**b**) represents the drought group, (**c**) represents the salt group, and (**d**) represents the ABA group.

**Figure 10 ijms-26-06289-f010:**
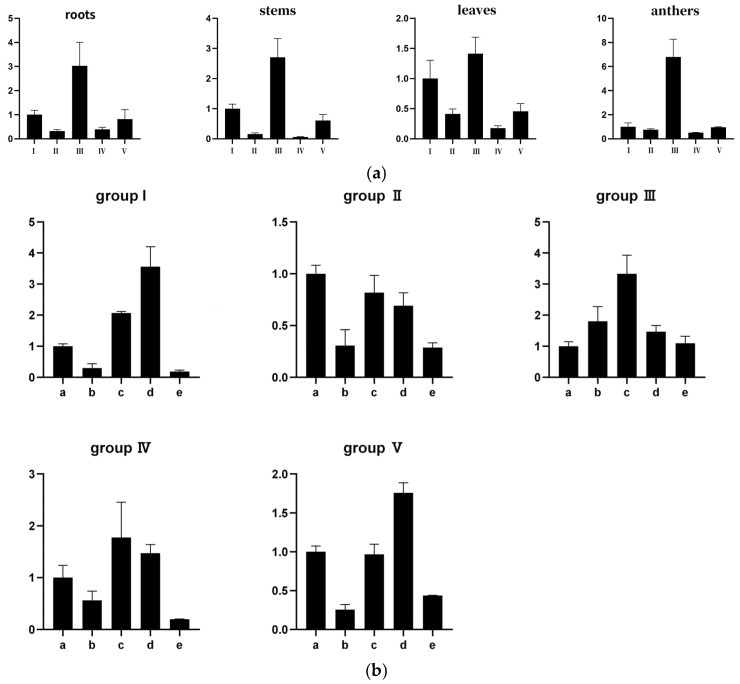
Expression of wheat SRS gene family under different tissue, abiotic stress and hormone treatment. “I” to “V” for “Group I” to “Group V” (**a**) represents the expression levels of family members in different tissues group. (**b**) represents the responses of family members to different stress conditions. Group I: *TaSRS8*. Group II: *TaSRS14*. Group III: *TaSRS15*. Group IV: *TaSRS4*. Group V: *TaSRS3*. qRT-PCR analysis of *TaSRS* in response to abiotic stress. “a” represents no coercion and is the control group. “b” stands for 6 h of cryogenic treatment. “c” represents 0.2 M Nacl treatment for 6 h. “d” represents 0.1 M ABA treatment for 6 h. “e” stands for 20% PEG treatment for 6 h. Group I: *TaSRS8*. Group II: *TaSRS14*. Group III: *TaSRS15*. Group IV: *TaSRS4*. Group V: *TaSRS3*. The Y-axis represents the relative expression level of the *TaSRS* gene, and the expression level of the control group is set to 1. The expression levels were calculated using the 2^−ΔΔCT^ method.

**Table 1 ijms-26-06289-t001:** *TaSRS* genes information identified in the wheat genome.

Gene Name	Gene ID	Length (aa)	Length (bp)	Molecular Weight	Isoelectric Point	Exon Number	Group
*TaSRS1*	TraesCS1A02G241400	314	945	32,544.10	8.89	3	V
*TaSRS2*	TraesCS1B02G253200	313	942	32,580.13	8.89	3	V
*TaSRS3*	TraesCS1D02G241300	313	942	32,449.04	8.89	3	V
*TaSRS4*	TraesCS3A02G491500	247	744	25,092.11	8.43	2	IV
*TaSRS5*	TraesCS3B02G552400	352	1059	35,484.59	7.72	2	IV
*TaSRS6*	TraesCS3D02G497300	348	1047	35,259.38	7.70	2	IV
*TaSRS7*	TraesCS5A02G319200	330	993	33,920.42	8.96	2	I
*TaSRS8*	TraesCS5B02G319600	330	993	33,924.41	8.96	2	I
*TaSRS9*	TraesCS5D02G325400	330	993	33,920.42	8.96	2	I
*TaSRS10*	TraesCS7A02G270900	315	948	32,500.40	8.16	2	II
*TaSRS11*	TraesCS7A02G406300	346	1041	34,382.91	8.68	2	III
*TaSRS12*	TraesCS7B02G169200	335	1008	34,636.99	7.71	3	II
*TaSRS13*	TraesCS7B02G306300	346	1041	34,468.96	8.41	2	III
*TaSRS14*	TraesCS7D02G271400	315	948	32,386.30	8.16	2	II
*TaSRS15*	TraesCS7D02G400000	346	1041	34,428.93	8.68	2	III

**Table 2 ijms-26-06289-t002:** Primers for quantitative real-time PCR of wheat SRS gene family.

Name	Gene	Forward Primer Sequences (5′→3′)	Reverse Primer Sequences (5′→3′)
group I	*TaSRS7*, *TaSRS8*, *TaSRS9*	CTGCTCCACCCACGTCAAGTCCA	AGACGCACGCATCGGAACACC
group II	*TaSRS10*, *TaSRS12*, *TaSRS14*	CAGACCACCGTCAGCATCG	GTGGCCGTGGAAGAATGG
group III	*TaSRS11*, *TaSRS13*, *TaSRS15*	GTCGTGTTGGGCTTCTCGC	GACCCAGGTGCTCTTGACG
group IV	*TaSRS4*, *TaSRS5*, *TaSRS6*	GCACGTCAAGAGCACCTGGGTC	ACGCCCTGGTCGTAGAGGAAGC
group V	*TaSRS1*, *TaSRS2*, *TaSRS3*	CGCCATCCAGTTCTGGCAATC	TCCTGACGTGCGTGGTGCA

## Data Availability

The datasets used and/or analyzed during the current study are available from the corresponding author on reasonable request.

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
