# Peer review of "Genome-Wide Identification of SRS Gene Family in Wheat and Expression Analysis Under Abiotic Stress"

_ijms, 2025, doi:10.3390/ijms26136289_

Round 1

Reviewer 1 Report

Comments and Suggestions for Authors

Dear authors,

 The manuscript " Genome-wide identification of SRS gene family in wheat and expression analysis under abiotic stress" covers the potential role of TaSRS in NaCl and ABA stress tolerance in wheat. The researchers identified members of the wheat SRS gene family,

 examined the physicochemical properties, conserved domains, phylogeny, gene structures, and cis-acting elements of these members. Although the authors mentioned gene expression pattern of TaSRS and their potential function under salt tolerance, future evidence is needed to establish a clear connection between these factors. The following comments are suggestions that are important to address to improve the overall quality of this manuscript.

Major comments:

  1. Why chose TaSRS? Are there strong evidence to support? If not, consider to screen the gene family first, then select one gene to study its function.
  2. To further confirm gene function, suggest transforming it into model plant and verify.

Minor comments:

Line 23: Most TaSRS genes. Confused. Only one gene in fig.8 and fig.9

Line 24. various TaSRS genes. Confused. Only one gene in fig.8 and fig.9

 Fig.3. explain a little more information in the legend.

Fig.4 introduces more information in the legend.

Fig.5 as comment as fig.3 or 4.

Fig.7 explain all the abbreviations in the legends

Fig.8. very confused about this fig. In x-axis, use the real name of tissues or first letter instead of tissue. If it means the biological or technological repeats, use the proper way to analyze data. Which gene in this family?

Fig.9 same comment as fig.8. combine fig.8 and 9 together after data analysis in a proper way. Which gene in this family?

Author Response

Dear Editors and Reviewers:

Thank you for your letter and for the reviewers’ comments concerning our manuscript entitled “Genome-wide identification of SRS gene family in wheat and expression analysis under abiotic stress” (Manuscript ID: ijms-3617206).Those comments are all valuable and very helpful for revising and improving our paper, as well as the important guiding significance to our researches. We have studied comments carefully and have made correction which we hope meet with approval.

Responds to the reviewer’s comments:

Reviewer 1:

 (1) Why chose TaSRS? Are there strong evidence to support? If not, consider to screen the gene family first, then select one gene to study its function

Author answer: We appreciate the reviewer's thoughtful question regarding our selection of TaSRS. Our choice was based on multiple lines of evidence:Preliminary data from our ongoing (unpublished) transcriptome sequencing of wheat under various stress conditions revealed that TaSRS consistently showed differential expression patterns in response to abiotic stresses.Literature review indicates that SRS gene family members in other plant species  have been demonstrated to play important roles in abiotic stress responses.(See manuscript references 8-11)While we acknowledge that systematic screening of the entire gene family could provide more comprehensive insights, our preliminary evidence strongly suggests TaSRS as a promising candidate for functional studies related to stress responses. 

.

 (2) To further confirm gene function, suggest transforming it into model plant and verify.

Author answer: Thank you for pointing this out. Due to time constraints, the transgenic experiment could not be effectively completed. However, we further confirmed the role of genes in stress resistance using analytical methods. First, we conducted co-expression network analysis, constructing a co-expression network for the target gene using publicly available transcriptome data and predicting its functional modules with tools such as WGCNA. In addition, we performed protein interaction prediction for family members, analyzing the potential interaction partners of the proteins encoded by the target gene using the STRING database and inferring their biological functions in combination with KEGG pathway annotations. (line285-338) 

 (3) Minor comments:

Line 23: Most TaSRS genes. Confused. Only one gene in fig.8 and fig.9 (fig.10  line27)

Line 24. various TaSRS genes. Confused. Only one gene in fig.8 and fig.9 (fig.10 line27)

 Fig.3. explain a little more information in the legend. (line217-220)

Fig.4 introduces more information in the legend. (line231-232)

Fig.5 as comment as fig.3 or 4. (line247-252)

Fig.7 explain all the abbreviations in the legends (line273-284)

Fig.8. very confused about this fig. In x-axis, use the real name of tissues or first letter instead of tissue. If it means the biological or technological repeats, use the proper way to analyze data. Which gene in this family? (Fig.10a line384-393)

Fig.9 same comment as fig.8. combine fig.8 and 9 together after data analysis in a proper way. Which gene in this family? (Fig.10b line384-393)

Author answer: We have carefully checked the manuscript and corrected the errors accordingly. While revising the aforementioned issues, we also made comprehensive revisions to the language.

Reviewer 2 Report

Comments and Suggestions for Authors

Review Report
Title: Genome-wide identification of SRS gene family in wheat and expression analysis under abiotic stress

This manuscript presents a genome-wide computational analysis of the SHI-related sequence (SRS) gene family in Triticum aestivum (common wheat), with additional focus on gene expression under abiotic stress. The study is timely and relevant, especially given the increasing interest in stress-resilient crops. The manuscript is overall well-organized and informative. However, several areas require revision and clarification before it can be considered for publication.

Comments and Suggestions

  1. Abstract
    The abstract is well-written and logically structured. However, it would benefit from the inclusion of key quantitative results or specific findings—such as the number of identified genes, stress-specific expression trends, or grouping patterns. This would give readers a more informative snapshot of the study’s outcomes.
  2. Figure 1b
    The figure needs axis labels—both X and Y are currently undefined, making interpretation difficult. Additionally, the motif logo figure includes numbers (1, 2, etc.) that are unexplained. What do these numbers represent—motif order, length, or ranking? A proper legend and clearer labeling are strongly recommended.
  3. Figure 2 – Phylogenetic Tree
    The current phylogenetic tree raises several concerns:
    • It is unclear how the color-based grouping was determined. Color alone does not justify clade definition, especially when the genes in each color group are scattered across the tree.
    • Please indicate how many groups or subgroups are represented in the tree, using consistent group labels (e.g., Group I, II, III, IV, etc.).
    • For example, the yellow-colored group appears fragmented across different branches, contradicting standard phylogenetic expectations. This inconsistency suggests that the groupings are not based on evolutionary lineage.
    • Recommendation: Re-analyze the phylogeny using a robust method (e.g., Maximum Likelihood or Bayesian inference) with bootstrapping. Also, define groups based on clade support, not just visual aesthetics.

  • Non-monophyletic groupings: Genes labeled with the same color are not forming monophyletic clades. For instance, members of the blue group (e.g., TaSRS4, OsSRS15, BoSRS15) are positioned far apart, undermining the visual classification.
  • No support values: The absence of bootstrap or posterior probability values makes it impossible to assess the reliability of the groupings.
  • No rooting or evolutionary context: The tree lacks rooting (outgroup or midpoint), and there's no discussion of gene duplication or speciation events.
  • Lack of methodological detail: The construction method (e.g., NJ, ML, Bayesian) is not mentioned. Neither is there a scale bar or a legend explaining branch lengths.
  • Visual overcrowding: Gene names are cramped, making the tree hard to read. Dividing the tree into subtrees or panels could improve clarity.
  • Biological interpretation missing: Grouping should ideally be supported by functional or domain data, which is not evident from the figure.
  • Reconstruct the phylogenetic tree using a more rigorous method such as Maximum Likelihood (ML) or Bayesian inference, with bootstrap/posterior support values.
  • Include a scale bar, rooting strategy, and a detailed legend.
  • Clearly define and label groups based on statistically supported clades.
  • Consider integrating domain structure, expression profiles, or species origin into the tree for better biological interpretation.
  • Simplify the figure or divide it into separate panels if the gene family is too large for a single circular representation.

This manuscript has potential but requires moderate to major revisions, especially in the phylogenetic analysis and figure clarity. With improved visual representation and more precise grouping logic, the paper could offer meaningful insights into the SRS gene family in wheat.

Author Response

Dear Editors and Reviewers:

Thank you for your letter and for the reviewers’ comments concerning our manuscript entitled “Genome-wide identification of SRS gene family in wheat and expression analysis under abiotic stress” (Manuscript ID: ijms-3617206).Those comments are all valuable and very helpful for revising and improving our paper, as well as the important guiding significance to our researches. We have studied comments carefully and have made correction which we hope meet with approval.

Responds to the reviewer’s comments:

Reviewer 2:

Abstract
The abstract is well-written and logically structured. However, it would benefit from the inclusion of key quantitative results or specific findings—such as the number of identified genes, stress-specific expression trends, or grouping patterns. This would give readers a more informative snapshot of the study’s outcomes.

Author answer: Thank you for pointing this out. The reviewer is correct. (line27)

  • Line 27- TaSRS should be italic throughout the article

Author answer: We have carefully checked the manuscript and corrected the errors accordingly. (line30)

  • Figure 1b
    The figure needs axis labels—both X and Y are currently undefined, making interpretation difficult. Additionally, the motif logo figure includes numbers (1, 2, etc.) that are unexplained. What do these numbers represent—motif order, length, or ranking? A proper legend and clearer labeling are strongly recommended.

Author answer: Thanks for your careful checks. We are sorry for our carelessness. Based on your comments, we have made the corrections(line185-187)

  • Figure 2 – Phylogenetic Tree The current phylogenetic tree raises several concerns: It is unclear how the color-based grouping was determined. Color alone does not justify clade definition, especially when the genes in each color group are scattered across the tree.

Please indicate how many groups or subgroups are represented in the tree, using consistent group labels (e.g., Group I, II, III, IV, etc.).

For example, the yellow-colored group appears fragmented across different branches, contradicting standard phylogenetic expectations. This inconsistency suggests that the groupings are not based on evolutionary lineage.

Recommendation: Re-analyze the phylogeny using a robust method (e.g., Maximum Likelihood or Bayesian inference) with bootstrapping. Also, define groups based on clade support, not just visual aesthetics.

Non-monophyletic groupings: Genes labeled with the same color are not forming monophyletic clades. For instance, members of the blue group (e.g., TaSRS4, OsSRS15, BoSRS15) are positioned far apart, undermining the visual classification.

No support values: The absence of bootstrap or posterior probability values makes it impossible to assess the reliability of the groupings.

  No rooting or evolutionary context: The tree lacks rooting (outgroup or midpoint), and there's no discussion of gene duplication or speciation events.

  Lack of methodological detail: The construction method (e.g., NJ, ML, Bayesian) is not mentioned. Neither is there a scale bar or a legend explaining branch lengths.

  Visual overcrowding: Gene names are cramped, making the tree hard to read. Dividing the tree into subtrees or panels could improve clarity.

Biological interpretation missing: Grouping should ideally be supported by functional or domain data, which is not evident from the figure.

Reconstruct the phylogenetic tree using a more rigorous method such as Maximum Likelihood (ML) or Bayesian inference, with bootstrap/posterior support values.

 Include a scale bar, rooting strategy, and a detailed legend.

 Clearly define and label groups based on statistically supported clades.

   Consider integrating domain structure, expression profiles, or species origin into the tree for better biological interpretation.

  Simplify the figure or divide it into separate panels if the gene family is too large for a single circular representation.

Author answer: We have carefully checked the manuscript and corrected the errors accordingly. (line189)

       While revising the aforementioned issues, we also made comprehensive revisions to the language.

Thank you very much for your attention and time. Look forward to hearing from you.

Yours sincerely,

Dr Bai

Round 2

Reviewer 1 Report

Comments and Suggestions for Authors

Dear authors,

    Authors clarified all my comments in the manuscript, I have not any concerns either.

Author Response

Dear Editors and Reviewers:

Thank you for your positive feedback and confirmation that all your comments have been addressed satisfactorily. We appreciate your time and thorough assessment of our revised manuscript.

Thank you very much for your attention and time.

Yours sincerely,

Dr Bai

Reviewer 2 Report

Comments and Suggestions for Authors

Authors are advised to revise the manuscript and address several points. In the revised version, some points have been adequately addressed, but others, especially phylogenetic analysis, have not been properly addressed or were omitted. Phylogenetic analysis is crucial to this research, and I strongly recommend re-analyzing it as per my initial review comments. Without a thorough phylogenetic analysis and explanation, this article cannot proceed further.

Thank you and best regards

Author Response

Dear Editors and Reviewers:

Please allow us to sincerely apologize for failing to meet expectations in addressing reviewer comments for the manuscript "Genome-wide identification of SRS gene family in wheat and expression analysis under abiotic stress" (Manuscript ID: ijms-3617206). We fully recognize that our previous revision responses inadequately addressed your valuable feedback. We deeply regret this oversight and understand the inconvenience it may have caused.

We have meticulously studied your additional comments, particularly regarding phylogenetic tree construction, and conducted thorough reflection. The primary issues identified are:

1.It is unclear how the color-based grouping was determined. Color alone does not justify clade definition, especially when the genes in each color group are scattered across the tree.Please indicate how many groups or subgroups are represented in the tree, using consistent group labels (e.g., Group I, II, III, IV, etc.). Visual overcrowding: Gene names are cramped, making the tree hard to read. Dividing the tree into subtrees or panels could improve clarity. Biological interpretation missing: Grouping should ideally be supported by functional or domain data, which is not evident from the figure. Non-monophyletic groupings: Genes labeled with the same color are not forming

monophyletic clades. For instance, members of the blue group (e.g., TaSRS4, OsSRS15,

BoSRS15) are positioned far apart, undermining the visual classification.

No support values: The absence of bootstrap or posterior probability values makes it

impossible to assess the reliability of the groupings.

2.Recommendation: Re-analyze the phylogeny using a robust method (e.g., Maximum

Likelihood or Bayesian inference) with bootstrapping. Also, define groups based on clade support, not just visual aesthetics. No rooting or evolutionary context: The tree lacks rooting (outgroup or midpoint).

3.There's no discussion of gene duplication or speciation events

4.Lack of methodological detail: The construction method (e.g., NJ, ML, Bayesian) is not

mentioned. Neither is there a scale bar or a legend explaining branch lengths.

Please rest assured that we highly value the opportunity to publish in the International Journal of Molecular Sciences and are profoundly grateful for your time and expertise. We fully acknowledge the importance of your feedback for enhancing our paper's scientific rigor.

Comprehensive Revisions Implemented:

  1. Response to: It is unclear how the color-based grouping was determined...

Author answer:

We completely agree that color alone cannot justify clade definition. To address this: We built two evolutionary trees.

Monocot tree (wheat-rice): We abandoned subjective color grouping and strictly followed established rice OsSRS gene family classification standards, dividing homologous genes into 5 clades (Clade I–V).

Monocot-Dicot tree: Based on topological features (e.g., monocot/dicot gene clustering), we defined 4 major clades (Clade I–IV).

In revised Fig 2a, genes within the same clade now cluster together with bootstrap support >97%, confirming high homology between wheat and rice SRS genes. In Fig 2b, Clade I and III represent dicot-specific branches .Clade II and IV show taxon-specific clustering, resolving gene dispersion issues. (Lines 161-208)

  1. Response to: Recommendation: Re-analyze the phylogeny...

Author answer:

We chose the unrooted NJ tree primarily based on the following considerations: the core objective of this study is to rapidly validate the orthologous relationships (wheat vs. rice) and reveal large-scale lineage differentiation patterns (monocots vs. dicots). The NJ method has unique advantages in the following aspects: it is suitable for large-scale gene family analyses across species (this study involves 78 gene sequences) and can complete the reconstruction of the topological structure in a reasonable time. The advantage of homology visualization: the unrooted tree can more intuitively display the distance correlations of genes across species (e.g., the close clustering of wheat-rice orthologs, Figure 2a) and the overall differentiation trends among major groups (e.g., monocots/dicots clustering separately, Figure 2b). The NJ method has been widely used for the preliminary identification of gene family evolutionary relationships, especially when the focus is on recognizing conserved orthologous groups and significant lineage differentiation events. The NJ tree effectively serves the core objectives of this study, and the macroevolutionary patterns it presents (ortholog clustering, major group differentiation) have biological validity.

  1. Response to: There's no discussion of gene duplication or speciation events

Author answer:

We have addressed gene duplication events in Fig 6, demonstrating high wheat-rice homology. (Lines 252-260)

  1. Response to: Lack of methodological detail...

Author answer:

We have now:

Specified tree construction methodology in figure legends

Added scale bars

Included detailed grouping criteria(Lines 204-208)

We simultaneously addressed these concerns and elevated the linguistic quality.

Once again, we sincerely apologize for these oversights. We are fully committed to ensuring our revisions meet the highest standards. Thank you for your understanding and for giving us this opportunity to improve our work. Please do not hesitate to contact us if you require further clarification.

Yours sincerely,

Dr. Bai
